# Nitrogen and Phosphorus of Plants Associated with Arbuscular and Ectomycorrhizas Are Differentially Influenced by Drought

**DOI:** 10.3390/plants11182429

**Published:** 2022-09-17

**Authors:** Manman Jing, Zhaoyong Shi, Mengge Zhang, Menghan Zhang, Xiaohui Wang

**Affiliations:** 1College of Agriculture, Henan University of Science and Technology, Luoyang 471023, China; 2Henan Engineering Research Center of Human Settlements, Luoyang 471023, China; 3Luoyang Key Laboratory of Symbiotic Microorganism and Green Development, Luoyang 471023, China

**Keywords:** drought stress, drought duration drought operation types, arbuscular mycorrhiza, ectomycorrhiza

## Abstract

Leaf nitrogen (N) and phosphorus (P) are the most important functional traits in plants which affect biogeochemical cycles. As the most widely observed plant–fungus mutualistic symbiosis, mycorrhiza plays a vital role in regulating plant growth. There are different types of mycorrhiza with various ecological functions in nature. Drought, as a frequent environmental stress, has been paid more and more attention due to its influence on plant growth. Numerous studies have confirmed that drought affects the concentration of N and P in plants, but few studies involve different mycorrhizal types of plants. In this study, the differences of N and P between arbuscular mycorrhizal (AM) and ectomycorrhizal (ECM) plants under different drought patterns, drought duration and cultivation conditions were explored based on a dataset by a meta-analysis. Drought stress (DS) showed negative effects on AM plant N (−7.15%) and AM plant P (−13.87%), and a positive effect on AM plant N:P ratio (+8.01%). Drought significantly increased N and the N:P ratio of ECM plants by 1.58% and 3.58%, respectively, and decreased P of ECM plants by −2.00%. Short-term drought (<30 d) reduces more N and P than long-term drought (<30 d) in AM plant species. The duration of drought did not change the N concentration of ECM plant N, while short-term drought reduced ECM plant P. The effects of N and P on DS also varied with different planting conditions and functional groups between AM and ECM plants. Therefore, mycorrhizal effects and stoichiometry of N and P play a key role in plant response to drought. So mycorrhizal effects should be considered when studying plant responses to drought stress.

## 1. Introduction

Mycorrhiza is a symbiosis formed by mycorrhizal fungi in soil and the roots of higher plants [1], which is one of the most important biological interactions [2]. The acquisition of plant nutrients almost always depends on the symbiotic relationship with mycorrhiza [3]. Mycorrhizal fungi promotes the absorption and utilization of water and mineral elements [4], significantly reduced soil N and P loss [5], and also affects the mineralization of N and P [6].

Arbuscular mycorrhizal fungi and ectomycorrhizal fungi are the most widely distributed mycorrhizal types in the plant kingdom. area total of 80% of land plants establish a mutually beneficial symbiotic relationship with arbuscular mycorrhizal fungi [1,7], such as ectomycorrhizas; however, most are commonly symbiotic with woody plants that live in cold regions of the world. In recent decades, more and more studies have found that the ecological functions of AM and ECM are different. Chen et al. (2022) found that soil concentrations of soluble organic nitrogen and nitrate N in AM forest were higher than those in ECM forest, and the nitrogen cycling and transformation process of AM forest and ECM forest significantly differed [8]. Liu et al. (2020) believed that AM and ECM lead to plants having different stoichiometry of N and P [9]. Shi et al. (2021) found that the stoichiometry of nitrogen and phosphorus in legumes was closely related to mycorrhizal traits [10]. Cheeke et al. (2017) pointed out that various species have different types of mycorrhiza, and functional differences between AM and ECM will lead to differences in N and P between plants [11].

N and P, the basic elements of life on Earth, are considered to be the main sources of plant minerals and organic nutrients [12,13]. N and P regulate photosynthesis and decomposition of litters [14], and participate in respiration of plants. The stoichiometry of N and P is the basic component of predicting ecological response to global change [15,16,17], which is an effective indicator of plant growth limitation in soil [18], absorption of N and P be affected by drought [19]. 

Drought stress (defined as water shortage in soil or atmosphere) is one of the important abiotic factors affecting plant physiological and biochemical processes [20,21]. Drought reduces stomatal conductance of plants, thus decreasing photosynthesis and transpiration rates [22]. Drought affects metabolism and substance synthesis in plants by affecting the acquisition, transport, distribution and storage of elements such as N and P [23]. He et al. (2014) used a meta-analysis to analyze the responses of plant N, P and N:P ratio to drought stress, and found that N and P had different responses to drought in various drought modes, drought duration and different plant groups, but this study ignored the role of mycorrhizal in drought [24]. 

Numerous studies have shown that mycorrhizae improve the drought resistance of plants. Zhu et al. (2012) pointed out that arbuscular mycorrhiza had an impact on corn growth, gas exchange, photosynthesis and transpiration under potted conditions. AM alleviates the toxicity of drought stress by improving the net photosynthesis rate and the transpiration rate of corn [25]. Augé (2004) proposed that AM, under drought conditions, improve water relationship, root water absorption rate and leaf water potential. Thus AM regulates the transpiration rate and enhances the photosynthesis rate [26]. ECM directly or indirectly improve drought tolerance of plants by promoting their absorption of water and minerals, improving photosynthesis, regulating osmotic substances and signal molecules [27,28,29]. Lin et al. (2013) pointed out that arbuscular mycorrhiza and ectomycorrhiza promote the absorption of nutrients by host plants, enhance the stress resistance of plants and help plants cope with stress [30]. 

All above studies have shown that mycorrhizae have certain effects on drought stress and nutrient absorption, However, are the effects of plant N and P in plants related to mycorrhizal type under drought? In this study, we investigated the effects of drought on different plants and proposed the following hypotheses: (1) The stoichiometric effects of drought on N, P and the N:P ratio of AM and ECM plants were different. (2) The responses of N, P and the N:P ratio of AM and ECM plants to different drought patterns were different. 

## 2. Results

### 2.1. Overall Differences in the Effects of DS on AM and ECM Plant N, P, and N:P Ratio

Analysis and comparison of 155 data results showed that drought stress had significantly different effects on the N, P, and N:P ratio of AM and ECM plants (Figure 1). The N, P, and N:P ratio effect sizes and their 95% CIs did not overlap with 0 for AM plants, indicating a significant DS effect. The 95% effect interval of ECM plants overlapped with 0, and there were no significant differences in ECM plant N, P and N:P ratio (*p* = 0.3187, *p* = 0.4249, *p* = 0.0531). These results indicated that the responses of N and P to drought stress were obviously different between AM and ECM plants (Figure 1).

### 2.2. Differences of DS Type and Manipulation Time on AM and ECM Plant N

The absorption of AM and ECM plant N responded differently to the type and duration of DS operation. For AM plants, the average effect size, in the constant-stressed type, was significantly lower than zero (a 0.1581% reduction on average, Figure 2a). Treatment time had an obvious effect on AM plant N absorption. The average effect size of AM plant N in the short-term (<90 d) treatment has a significant reduction (−0.1495%, Figure 2a), while there was no significant difference in the long-term (>90 d) treatment. The N ECM plants absorb varies most during drying–rewetting cycle type operations (+0.0844%, Figure 2a). As for the effect of treatment time, ECM and AM plants are affected the most under short-term treatment, but AM plant N are affected more than ECM plants.

### 2.3. Differences of DS Type and Manipulation Time on AM and ECM Plant P

The absorption of AM and ECM plant P has diverse differences in DS manipulation type and operation time. There was the greatest impact on AM plant P during operational treatments of the intermittent drying type (Figure 2b). The uptake of AM plant P was different during treatment time, with the greatest impact on P uptake during short-term treatments (+0.1852%, Figure 2b). ECM plants, in drying–rewetting cycle type, has the highest absorption effect on P (+0.2287%, Figure 2b). Short-term treatment had the greatest effect on P (+0.2234%, Figure 2b), and more than AM plants.

### 2.4. The Difference of DS Type and Manipulation Time on AM and ECM Plant N:P Ratio

For AM and ECM plant N:P ratio, the difference of operation time was distinct due to different operation types. For AM plants, there was no significant difference in the effects of type Ⅰ and Ⅱ, but the effects of type III on plant N:P ratio were extremely significant (+0.1061%, Figure 2c). The N:P ratio of AM plants was affected by Long-term drought treatment (Figure 2c). The opposite of AM plants, the N:P ratio of ECM plants changed significantly under short-term drought (+0.1831%, Figure 2c).

### 2.5. Different Functional Groups of AM and ECM Showed Different Responses of N, P and N:P Ratio to Drought

Drought had no effect on N in AM and ECM woody plants (Figure 3a). Drought significantly reduced P in AM woody plants, but there was no change in P in ECM woody plants (Figure 3a). The N:P ratio of AM woody plants increased significantly under drought conditions, while ECM woody plant N:P ratio is not affected by drought (Figure 3a). AM plants are further divided into herbaceous and woody plants. The responses of N, P and N:P ratio to drought were also different between AM woody plants and herbaceous plants. N and P of AM herbaceous plants decreased significantly, while N of AM woody plants did not change, and P decreased significantly. Thus, it can be seen that for AM plants, the change in N under drought conditions is mainly caused by herbaceous plants. (Figure 3b). When woody plants were further divided into trees and shrubs, it was found that drought stress had no effect on nitrogen of trees and shrubs in AM and ECM woody plants (Figure 4a). Tree phosphorus content in AM plants decreased significantly under drought stress, this indicates that the P of AM plants in woody plants will decrease under drought conditions mainly due to the effect of trees in woody plants. Drought had no effect on shrub phosphorus concentration in AM and ECM plants (Figure 4b). The N:P ratio in the tree of AM woody plants increased significantly under drought stress (Figure 4c), this corresponds to the result in Figure 3a.

### 2.6. Overall Difference and Distribution Range of N, P, and N:P Ratio Effects of Drought on AM and ECM Plants

In general, DS reduce the concentration of AM plant N by 7.15% on average, with a minimum value of −0.75 (Figure 5a). The average P of AM plants decreased by 13.87% (Figure 5c). The N:P ratio of AM plants also increased by 8.01% on average due to changes in N and P (Figure 5e). N of ECM plants increased by 1.58% on average, mainly concentrated in [−0.136, 0.176] (Figure 5b). P of ECM plants decreased by 2.00% on average, and the data in the figure were consistent with a normal distribution according to S, SS, K, and SK values, so the P concentration of ECM plants was mainly distributed in [−0.269, 0.229] (Figure 5d), which was much lower than that of AM plants. The N:P ratio of ECM plants also increased, with an average increase of 3.58% (Figure 5f). The P values shown in Figure 5a–f (*p* < 0.001; *p* = 0.468; *p* < 0.001; *p* = 0.565; *p* = 0.017; *p* = 0.174), drought had a significant effect on N, P, and the N:P ratio of AM plants but had no effect on N, P, and the N:P ratio of ECM plants, which was consistent with the results in Figure 1.

### 2.7. Different Effects of Pot or Field Treatment on AM and ECM Plant N, P, and N:P Ratio

We used two planting conditions altogether—pot and field. Drought has a great influence on N concentration in both field and pot cultivations conditions, but the effect of drought on N in field experiment is greater than that of pot cultivation experiment (−0.1269%, Figure 6a). The absorption of P by AM plants decreased in both field and pot experiments (by 0.1451% and 0.1317%, respectively, Figure 6b). In field experiment, the absorption of the N:P ratio was not significantly affected, but the absorption of the N:P ratio, in pot experiment, was positively affected (+0.0994%, Figure 6c). For ECM plants, field experiment had a negative effect on N absorption, but the effect was smaller than that of AM plants. The N absorption of ECM plants had a positive effect in Pot experiment, and more than AM plants, but P does not change (Figure 6a). There was no significant difference in the effect of ECM plants on P in field and pot experiments (Figure 6b). The N:P ratio of ECM plants was improved in pot experiment (+0.1092%, Figure 6c), which was significantly higher than that under field conditions (a decrease of 0.0538, Figure 6c). In addition, the effect of ECM plant N:P ratio was greater than that of AM plants in both field and pot conditions.

### 2.8. Effects of Other Factors on N, P, and N:P Ratio of AM and ECM Plants

In addition to drought patterns, drought time also analyzed whether species distribution and planting conditions affected plant N, P, and N:P ratio. The results showed that planting conditions also had significant effects on plant nitrogen (*p* = 0.01028 < 0.05). Plant N, P, and N:P ratio were not affected by species distribution and their cross-pollination.

## 3. Materials and Methods

### 3.1. Data Compilation

In this study, a large number of published works were consulted and mycorrhizal types of plants in the database were determined by referring to Hempel et al. Wang B et al. and Harley et al. [31,32,33]. The plants were divided into AM and ECM mycorrhizal types. There were 103 AM and 52 ECM plants. Of the 155 plants, 67 were herbaceous and 88 were woody. This study measured the differences of N and P content between AM and ECM plants under different drought patterns and duration. In the study, the DS and control treatment started with the same soil type and plant species, and were conducted under equal spatial and temporal scales. DS in the studies was achieved by manipulating soil water content (SWC) in controlled-environment facility (potting experiment). We noted location, drought treatment methods, plant species and functional groups, treatment time and the response of variables. We calculated the standard deviation and standard error of sample size for field and pot. Drought stress in the field and pot experiments was implemented using different time scales (from 1 days to several years) and frequencies. We focused on the water deficit caused by DS operation type and drought duration. The drought operation was divided into three types (Figure 7): constant-stressed type—the control group and DS treatment were kept at a constant SWC through the duration of the experiment (with lower SWC for the DS treatment, type Ⅰ). The drying and rewetting cycling type both control and drought treatments underwent drying and rewetting cycles, but the drying and rewetting cycle for the DS treatment had overall lower SWC (type Ⅱ). Intermittent drying type—moisture content was reduced at a specific stage (type III). The duration of DS was grouped into two categories: short-term treatment (0–30 d) and long-term treatment (>90 d). On this basis, we analyzed the effects of two dominant mycorrhizal types, AM and ECM, on N and P uptake in plants under different drought patterns and different drought periods (Appendix A).

### 3.2. Data Analysis

Data were expressed as the mean value ± standard error. SPSS 19.0 software was used to for statistical analysis of the effect size frequency of N, P, and the N:P ratio of AM and ECM plants. For plant and soil parameters, we used the natural log of the response ratio as a metric of the effect size, Log_e_ R = Log_e_ (X_d_/X_c_) = Log_e_ (X_d_) – Log_e_ (X_c_), where X_d_ and X_c_ are the mean values of drought stress and control treatment, respectively. If log_e_ R = 0, DS processing is invalid.

The variance of log_e_ R was calculated using the following formula:v = S^2^_t_/ntX^2^_d_ + S^2^_c_ncX^2^_c_(1)

S_t_ and S_C_ represent SDs of the treatment group and control group, respectively. In addition, nt and nc were the sample sizes of the treatment group and the control group, respectively.

The random effects model in Meta Win 2.1 was used to calculate the average effect size, determine the effect size and generate a 95% confidence interval (CI). Drought stress treatments were considered significant if 95% CI did not overlap with 0. The logarithmic mean effect size was inversely transformed and the DS effect was reported as a percentage change compared to the control group. Using more than one observation in one study may overexpress the effects of studies with a large number of observations. To test if this was the case, we randomly selected one observation from each study and performed the same analysis on only the selected observations. The average effect size calculated for this selected database was similar to the average effect size for the entire dataset, indicating that there was no overreach of the effects for a particular study. We used Q-Inter statistical analysis to evaluate whether there were differences in the effects of DS operation type (I-IV), DS duration type (short, medium and long term) and experiment type (field and pot experiment) on AM and ECM plants. When P_random_ < 0.05, the difference is considered significant.

## 4. Discussion

AM and ECM exist widely in various ecological environments [1]. In recent decades, the role of AM and ECM in drought response has received increasing attention. Averill et al. (2019) conducted nutrient analysis on global plants with different mycorrhizal types and found that the acquisition strategies of N and P nutrients for AM and ECM plants differed globally [34]. Chen et al. (2022) compared and analyzed the difference of soil nitrogen status between AM and ECM forest types, indicating that the forest with AM species dominant had a faster soil N cycle and a higher net nitrification rate and nitrate N concentration than the forest with ECM species dominant [8]. Zhang et al. (2018) pointed out that on a global scale, the absorption of N and P by AM and ECM trees showed differences with the change in climate regions [35]. Thus, mycorrhiza plays a vital role in the ecosystem. In this study, the effect of mycorrhizas was considered when discussing the responses of plant N and P to drought stress. The results showed that the N and P of AM and ECM plants differences between N and P response to drought. Many studies have proved that various mycorrhizal types have different effects on plant growth, which is consistent with our conclusion. Therefore, mycorrhizal types of plants should be considered when studying the effects of drought on plants.

Our results showed that drought reduced the N concentration of AM plants but not ECM plants, which may be caused by the difference in N absorption between AM and ECM plants [36]. ECM mainly helps plants absorb N, while AM mainly helps plants absorb P, which may also reveal the difference of P between the two mycorrhizal types. The difference of N or P in AM and ECM plants may also be caused by the different living environments of the two types of mycorrhizal plants, because AM plants tend to live in areas where N is abundant but P is deficient, while ECM plants tend to be distributed in areas where N is deficient. This leads to a possible greater uptake of N by ECM in plants [1,37,38,39,40,41]. The P of AM plant significantly decreased, while ECM plant P had no effect, which may also be caused by the above two reasons. The N:P ratio also changed, with the change in N and P of AM and ECM plants. Additionally, the N:P ratio of AM plants increased significantly, while that of ECM plants did not change.

The effects of drought on N, P, and N:P ratio were different between AM and ECM plants, which may be related to drought pattern and drought duration. The results in Figure 1 showed that the N concentration of AM plants decreased significantly under continuous drought, while that of ECM plants did not change, which may be because the negative effects of drought on AM fungi were mostly inhibited in extreme environmental conditions [42]. In extreme or long-term drought conditions, AM fungi were inhibited in absorption of water and N and P. The N of AM plants, under periodic drought conditions, did not change, but ECM plants increased significantly. The P of AM and ECM plants both decreased significantly under drying–rewetting cycle type, and AM plant P nor ECM plant P changed under other drought conditions. This difference may be due to the fact that AM fungi are more symbiotic with herbaceous plants, while ECM is more symbiotic with woody plants (Figure 3). Herbaceous plants have relatively shallow roots and mainly absorb water and nutrients from the soil surface [43]. Figure 2 also showed that N and P of AM and ECM plants decreased significantly under short-term drought (<30 d), while N of AM and ECM plants was not affected under long-term drought (>90 d). The P of AM and ECM plants also changes dramatically in the short-term drought. Similar to the research results of He et al. (2014), DS caused short-term decreased in plant N and P, but these effects were alleviated in the long term [24]. However, Deng et al. (2021) studied the response of soil C and N cycles to drought in forest, shrub and grassland through Meta analysis [44]. It was found that the higher the drought intensity and the longer the drought duration, the greater the drought effect. The results of this study are contrary to this, and the reasons need to be further explored.

Further analysis of AM and ECM plants (Figure 3) showed that 33 woody plants and 70 herbaceous plants were found in AM plants, and all ECM plants were woody plants. Drought has no effect on N of AM and ECM woody plants, while drought has a significant negative effect on N of AM herbaceous plants. Therefore, it can be seen that the N of AM plants decreases significantly because of herbaceous plants. This may be because the N and P concentration are different between herbaceous and woody plants [45]. The N and P concentration of herbaceous plants are higher than those of woody plants, so the changes in the N and P concentration of herbaceous are greater than those of woody plants under drought conditions, which further confirms the results in Figure 1. P of AM woody and herbaceous plants decreased significantly, while P of ECM plants did not change. By further subdividing woody plants (Figure 4) and comparing trees and shrubs in woody plants, it was found that trees and shrubs in AM and ECM woody plants had different effects on N and P under drought conditions. In this study, we found that the responses of AM and ECM plant N and P to drought stress were different due to the duration categories (long term and short term), type of functional group (trees, shrubs, and grassland) and experiment types (field experiment and pot experiment, Figure 6).

Fu et al. (2022) showed that AM fungal communities were very sensitive to extreme drought [46]. The results of this study also showed that ECM plants had stronger adaptability than AM plants under drought conditions (Figure 1 and Figure 5). However, there were also studies that proved that AM plants had stronger drought tolerance than ECM plants [27,47,48]. There may be two reasons for the opposite results. One is caused by data variation. There are few research data and there may be large variation. Secondly, the planting conditions and species distribution were ignored. Our study did show that planting conditions would have an impact on plant N (Table 1), but it was not entirely caused by these indicators, and some of our other indicators may also have an impact. As a result, the specific reasons for the differences still need to be further explored.

## 5. Conclusions

The effects of drought on plant N and P are closely related to mycorrhizal types. The N, P and N:P ratio of AM plants were significantly affected by drought, and ECM far less so. The N, P and N:P ratio of AM plants were more sensitive to drought than ECM plants. The responses of AM and ECM plant N, P, and N:P ratio to drought stress were related to duration categories, functional group type and experiment type. In conclusion, these results revealed the role of mycorrhizas in plant response to drought and provided data support for future exploration of AM and ECM plants in response to climate change.

## Figures and Tables

**Figure 1 plants-11-02429-f001:**
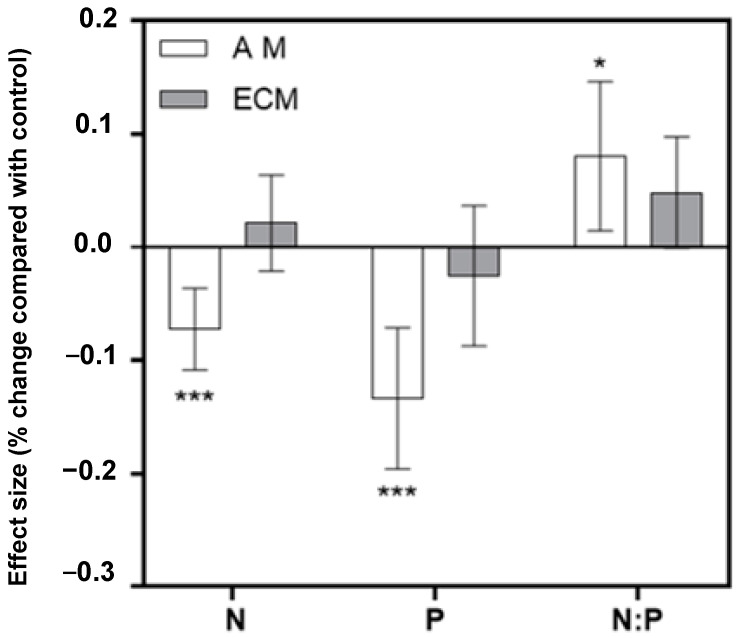
The error bar represents a 95% bootstrap confidence interval (CIs). If log_e_ R 95% CI did not overlap with zero, the effect of drought stress was considered significant. Here “*” means significant difference, and “***” means very significant difference.

**Figure 2 plants-11-02429-f002:**
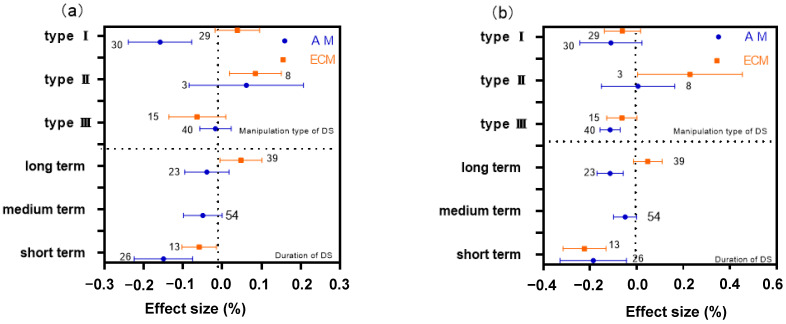
The mean effects of drought on arbuscular mycorrhizal N (**a**), P (**b**), and N:P ratio (**c**) and ectomycorrhizal N (**a**), P (**b**), and N:P ratio (**c**) were classified according to the type and duration of drought stress (DS). Type I, constant-stressed type; type II, drying and rewetting cycle type; type III, intermittent drying type. Short term, <30 d; medium term, 31 to 90 days; long term, >90 days. The error bar represents a 95% bootstrap confidence interval (CI). If the 95% CI of effect size did not overlap with zero, drought stress was considered to be the effect. The number of observations for each category is shown next to the error bar.

**Figure 3 plants-11-02429-f003:**
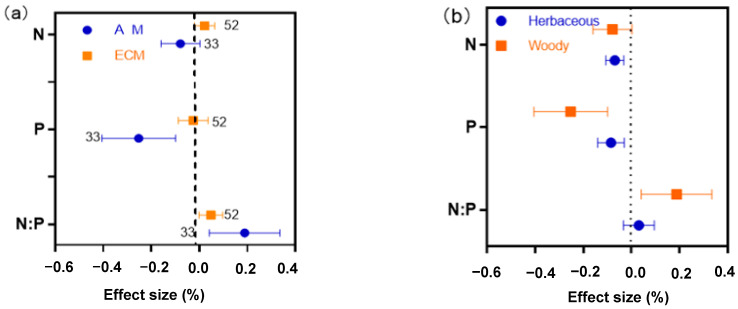
Effect of drought on woody plant N, P, and N:P ratio in AM and ECM plants (**a**), and N, P, and N:P ratio in herbaceous and woody AM plants (**b**).

**Figure 4 plants-11-02429-f004:**
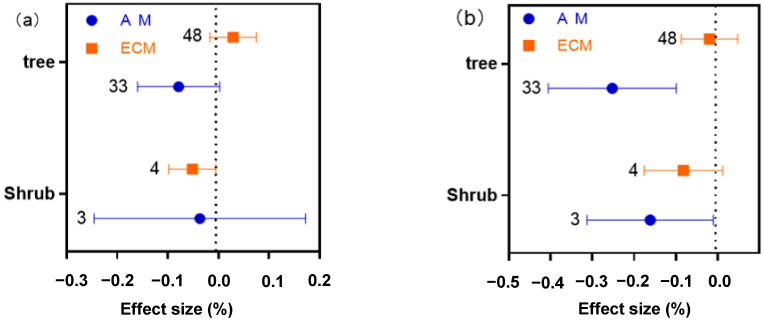
Effects of drought on AM and ECM plant N (**a**), P (**b**), and N:P ratio (**c**) were classified by tree and shrub.

**Figure 5 plants-11-02429-f005:**
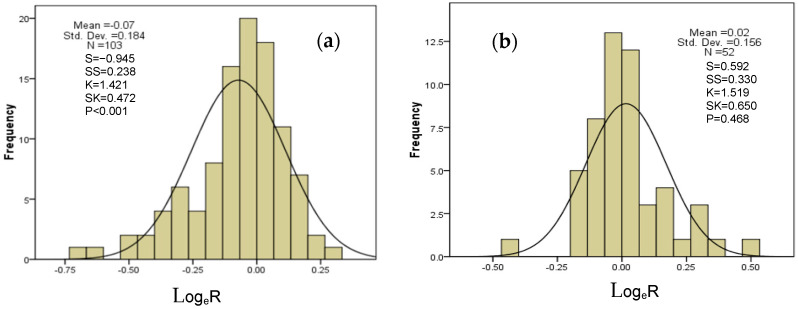
The frequency distribution of the effect size (natural log of the response ratio, Log_e_R) for arbuscular mycorrhizal plant N (**a**), ectomycorrhizal plant N (**b**), arbuscular mycorrhizal plant P (**c**), ectomycorrhizal plant P (**d**), arbuscular mycorrhizal plant N:P ratio (**e**), and ectomycorrhizal plant N:P ratio (**f**) to drought stress. Additionally, the mean effect size of drought stress on dendritic mycorrhizal plant N, P, and N:P ratio and ectomycorrhizal plant N, P, and N:P ratio. The solid curve in (**a**–**f**) is a Gaussian distribution matching the frequency data (n, number of observations). The error bars in (**f**) represent 95% bootstrapped confidence intervals (CIs). The effect of drought stress was considered significant if the 95% CI of loge R did not overlap with zero. In the figure, S refers to skewness, SS refers to the standard error of skewness, K refers to kurtosis, and SK refers to the standard error of kurtosis. If the |S − 0| / SS < 1.96, |K − 0|/SK < 1.96, the data follows normal distribution.

**Figure 6 plants-11-02429-f006:**
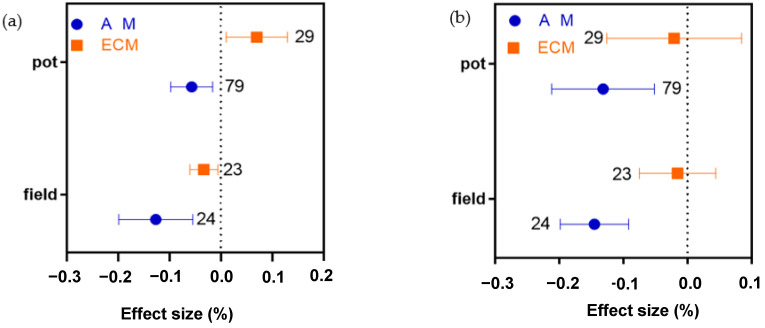
The average effect of drought on ectomycorrhizal plant N, (**a**) P, (**b**) and N:P ratio, (**c**) and arbuscular mycorrhizal plant N, (**a**) P, (**b**) and N:P ratio, (**c**) under field or pot experiment. The error bar represents a 95% bootstrap confidence interval (CI). If the 95% CI of effect size did not overlap with zero, drought stress was considered to be the effect. The number of observations for each category is shown next to the error bar.

**Figure 7 plants-11-02429-f007:**
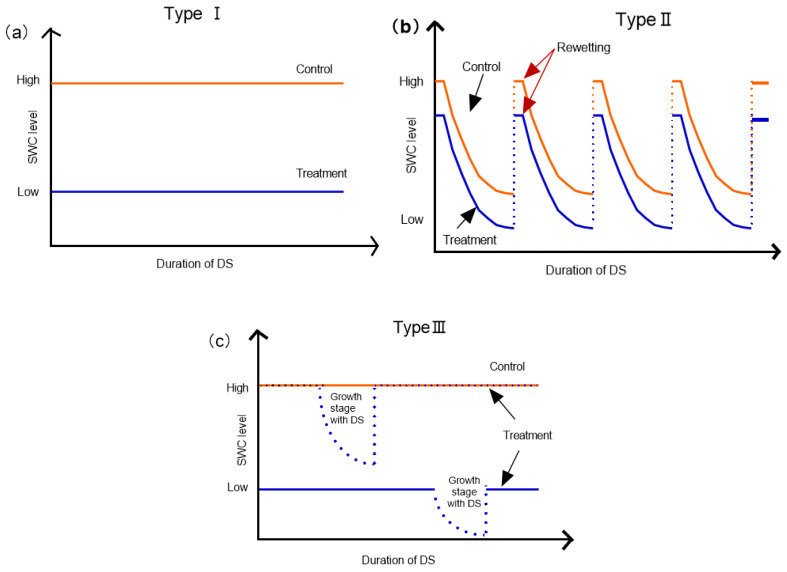
Drought stress (DS) manipulation type used in the meta-analysis. We classified DS manipulation treatments into the following three types: (**a**) type I, constant-stressed type—soil water content (SWC) in control and DS treatments is constant through the experiment with lower SWC in the DS treatment; (**b**) type II, drying–rewetting cycle type—control and DS treatments have an identical frequency of drying–rewetting cycles; however, the SWC content in the DS treatment was lower overall; (**c**) type III, intermittent drying type—the SWC in the DS treatment was the same as in the control treatment or at a lower level, but with a reduction in SWC during a specific growth stage period.

**Table 1 plants-11-02429-t001:** The effect of mycorrhiza type, planting condition, species distribution and their interaction on the N, P, and N:P ratio in AM and ECM plants based on linear mixed effect models planting conditions, species distribution and their interaction are fixed effects. Mycorrhizal type is random effects.

	Planting Conditions (PC)		Species Distribution (SD)		PC × SD	
	F	P	F	P	F	P
N	6.754	0.01	3.143	0.078	1.366	0.244
P	0	0.997	0.733	0.394	0.065	0.799
N:P	2.375	0.126	0.134	0.716	0.662	0.422

## Data Availability

The original contributions presented in the study are included in the article/Appendix A, further inquiries can be directed to the corresponding author/s.

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
