# Peer review of "Nitrogen and Phosphorus of Plants Associated with Arbuscular and Ectomycorrhizas Are Differentially Influenced by Drought"

_plants, 2022, doi:10.3390/plants11182429_

Round 1

Reviewer 1 Report

The manuscript by Jing et al. describes effects of drought stress on the concentration of leaf nitrogen and phosphorus in arbuscular mycorrhizal and ectomycorrhizal plants. The authors used meta-analysis to examine responses of nitrogen and phosphorus in arbuscular mycorrhizal and ectomycorrhizal plants to drought stress. The manuscript is overall well presented, and the materials are well collected to support the manuscript results. Therefore, I will only make some minor comments in the context of the results and discussion, which is too long. It could be written more precisely and accurately. My other concerns are,

Line 102: Please mention the name of used database.

Lines 103-106: Please summarized the outcomes of database in the tabular form as supportive information.

Lines 112-114: Again, showing this would benefit the readers. Please draw a supplementary table for this statement.

What does v and Xt stands for in the equation. Please explain this.

There are few typos in the text which can be corrected.

Reviewer 2 Report

This is a meta-analysis report to assess the differences of N and P between AM plants and ECM plants under different drought patterns, drought duration and cultivation conditions.

The manuscript is generally clear and interesting. However, the results seems to be inconsistent. The authors mentioned that data variation, planting conditions and species distribution might be the causes. Is it possible to analysis the influences of planting conditions and species distribution on the sensitivity of AM community or ECM community to drought? In order to make the work robust and with a greater impact on the scientific community, some detail show be addressed and the presentation of the results could be improved.

Round 2

Reviewer 1 Report

The manuscript entitled ‘Nitrogen and phosphorus of plants associated with arbuscular and ectomycorrhizas are differentially influenced by drought’ by Manman Jing and co-authors, have satisfactorily revised and improved the manuscript. Thank you for their efforts and quick response.

Author Response

Responses to Academic Editor

Line 50: “Pointed” - This word must begin with a lowercase letter.

Response: Accepted. Thank for your reminding. We have changed the capital P to a lowercase P.

Line 52: “ECM will lead to differences in nitrogen and phosphorus between plants” – should it be “N and P”?

Response: Accepted. We strongly agree with the reviewer, here nitrogen and phosphorus should be written as N and P.

Line 53: “N and P, as essential nutrients” -Please, avoid this repetition in this line.

Response: Accepted. Sorry, because of our carelessness, there were repeated expressions. We have adjusted and deleted “as essential nutrients”. The revised expression is provided as follows:

N and P, the basic elements of life on earth, are considered to be the main sources of plant minerals and organic nutrients.

Lines 70-71: “AM symbiotic bacteria” – It is not clear what do you mean? AM fungi?

Response: Accepted. We apologize that we did not provide a scientific presentation and we have revised it based on your suggestion.

Line 100: “response variables” – more correct should be “the response of variables”.

Response: Accepted. Thank for your reminding. We have changed “response variables” to “the response of variables”.

Line 103: “focusing” - more correct should be “focused”.

Response: Accepted. Expert is right. We have changed the focusing to focused.

Lines 113-114: Corrections of the text should be done.

Response: Accepted. We have carefully revised this sentence.

Our revised content is as follows:

On this basis, we analyzed the effects of two dominant mycorrhizal types AM and ECM on N and P uptake in plants under different drought patterns and different drought periods.

Lines 141-142, 157: Corrections of the text should be done.

Response: Accepted. Thanks to the reviewers for pointing out some of our mistakes.

Our revised content is as follows:

Lines 141-142: Drought stress treatment were considered significant if 95% CI did not overlap with 0.

Line 157: The N, P, N:P effects sizes and their 95% CIs did not overlap with 0 for AM plants, indicating significant DS effect.

Line 161: Please, start a new sentence here: “These results….”

Response: Accepted. Thank for your constructive comments. We start a new sentence at line 160 of the modified version.

Figure 3: The data for medium term (31 to 90 days) absent in the figure. They should be presented. Please add also the name and units of the x-axis.

Response: Accepted. We have appended the data of medium term (31 to 90 days) and added the x-axis name and unit, as shown in Figure 3.

Figures 4, 5, 6 and 7: Please add the name and units of the x-axis.

Response: Accepted. We have added the name and units of x-axis, as shown in figures 4, 5, 6 and 7.

Figure 6: I suggest to check distributions of the data in this figure for normality (to find difference from an expected normal distribution) using Skewness and Kurtosis criteria. Please show P values for these criteria to prove or reject normality. This should help you to discuss the obtained results.

Response: Accepted. Thank for your reminding. According to your suggestion, we used Skewness and Kurtosis criteria to check the normal distribution of the data in the graph, and annotated the P-value, which is shown in Figure 6 of the latest manuscript.

Line 292: Corrections of the text should be done.

Response: Accepted.

The modified content is as follows:

In this study, the effect of mycorrhizas was considered when discussing the responses of plant N and P to drought stress.

Lines 294-297: The meaning of this sentence is not clear.

Response: Accepted. We are sorry that we did not express our meaning clearly. We have made modifications as follows:

Many studies have proved that various mycorrhizal types have different effects on plant growth, which is consistent with our conclusion. Therefore, mycorrhizal types of plants should be considered when studying the effects of drought on plants.

Line 302: Corrections of the text should be done.

Response: Accepted. Thank you for your comments. We have corrected the sentence in line 304 of our modified version.

Lines 327-328: “long-term drought alleviates the negative effects of short-term drought on plants” – this speculation sounds confusing. This sentence should be corrected. Are there more explanations of the observed effect?

Response: Accepted. Sorry for our confusing expression, we have carefully

revised the content as follows:

Similar to the research results of He et al. (2014), DS caused short-term decreased in plant N and P, but these effects were alleviated in the long term.

Lines 345-348: “N and P, in pot and field conditions,” – it is not clear what do you mean. Please, make correction.

Response: Accepted. Thank for your constructive comments. We have carefully revised this sentence, in order to express our content clearly. The revised sentence is as follows:

In this study, we found that the responses of AM and ECM plants N and P to drought stress were different due to the duration categories (long-term, short-term), type of functional group (trees, shrubs, grassland) and experiment types (field experiment and pot experiment Figure. 6).

Line 352: This sentence sounds confusing. I suggest starting a new sentence after “(Figure. 5).” = But there were also…

Response: Accepted. We have modified it according to your suggestion, in line 353 of the modified version.

Lines 367-369: This sentence sounds confusing. It is not clear what do you mean?

Response: Accepted. Thanks the reviewer's constructive suggestions. Our revised statement is as follows:

In conclusion, these results revealed the role of mycorrhizas in plants response to drought and provided data support for future exploration of AM and ECM plants to in response to climate change.

Thank you for your comments on our manuscript again. We have tried our best to modify the manuscript. We will make further modification to improve the quality of the manuscript, if there are any problems.